UPDATE ARTICLE

# Dynamic collateral sensitivity profiles highlight opportunities and challenges for optimizing antibiotic treatments

**Jeff Maltas**[1]*, **Anh Huynh**[1], **Kevin B. Wood**[1,2]†

1 Department of Biophysics, University of Michigan, Ann Arbor, Michigan, United States of America,
2 Department of Physics, University of Michigan, Ann Arbor, Michigan, United States of America

† Deceased.
* jeff.maltas@gmail.com

## Abstract

As failure rates for traditional antimicrobial therapies escalate, recent focus has shifted to evolution-based therapies to slow resistance. Collateral sensitivity—the increased susceptibility to one drug associated with evolved resistance to a different drug—offers a potentially exploitable evolutionary constraint, but the manner in which collateral effects emerge over time is not well understood. Here, we use laboratory evolution in the opportunistic pathogen *Enterococcus faecalis* to phenotypically characterize collateral profiles through evolutionary time. Specifically, we measure collateral profiles for 400 strain-antibiotic combinations over the course of 4 evolutionary time points as strains are selected in increasing concentrations of antibiotic. We find that at a global level—when results from all drugs are combined—collateral resistance dominates during early phases of adaptation, when resistance to the selecting drug is lower, while collateral sensitivity becomes increasingly likely with further selection. At the level of individual populations; however, the trends are idiosyncratic; for example, the frequency of collateral sensitivity to ceftriaxone increases over time in isolates selected by linezolid but decreases in isolates selected by ciprofloxacin. We then show experimentally how dynamic collateral sensitivity relationships can lead to time-dependent dosing windows that depend on finely timed switching between drugs. Finally, we develop a stochastic mathematical model based on a Markov decision process consistent with observed dynamic collateral profiles to show measurements across time are required to optimally constrain antibiotic resistance.

## Introduction

Evolution of resistance continues to reduce the available set of drugs for successful treatment of bacterial infections, cancers, and viral infections [1–6]. As traditional maximum tolerable dose treatments fail at increasing rates, evolution-based treatments have emerged as a promising method to prolong the efficacy of current drugs or even reverse resistance. These treatments include drug cycling [7–10], harnessing spatial dynamics [11–14], cooperation [15–19], adaptive therapy [20–22], and judicious use of drug combinations [23–30]. More recently, there has been a growing focus on exploiting collateral sensitivity to slow or reverse evolution

**Data availability statement:** Relevant data are found within the paper and its Supporting information files or archived at https://doi.org/10.5281/zenodo.14064963.

**Funding:** This work was supported by NIH R35 GM124875 (to KBW). The funders had no role in study design, data collection and analysis, decision to publish, or preparation of the manuscript.

**Competing interests:** The authors have declared that no competing interests exist.

**Abbreviations:** BHI, brain heart infusion; CRO, ceftriaxone; CIP, ciprofloxacin; DOX, doxycycline; LZD, Linezolid; d-MDP, dynamic MDP; CR, frequency of resistance; CS, frequency of sensitivity; $IC_{50}$, half-maximal inhibitory dose; MDP, Markov decision process.

in bacteria and cancer [31–40]. Collateral evolution occurs when a population evolves resistance to a selecting drug and as a direct result exhibits increased or decreased resistance to a different drug.

While collateral sensitivity is a promising evolutionary therapy, a number of factors make its application to the clinic challenging; for example, collateral effects exhibit a high degree of heterogeneity [41,42], distinct collateral profiles arise from different selection pressures [43], collateral effects are often not repeatable [44], and many non-antibiotic environments can confer collateral sensitivity [45–50]. Despite these challenges, theoretical and laboratory studies have shown that control theoretic approaches may be used to counter, and even leverage, stochastic features of the evolutionary process to shape population outcomes [41,51]. Still, many fundamental questions about collateral sensitivity remain unanswered and are the focus of ongoing work. For example, the molecular mechanisms behind collateral sensitivity are known in relatively few cases [37,52], and it is unclear the extent to which collateral profiles are conserved across diverse species [53]. In addition, collateral drug pairs are difficult to identify in clinical settings, despite notable recent progress [54,55], and somewhat surprisingly, little is known about how collateral profiles change under continued selection, with much of the work performed only recently in other organisms such as cancer [38,56].

In this work, we sought to understand how collateral effects change over time in bacteria exposed to increasing antibiotic selection and, in turn, how these potentially dynamic collateral sensitivity profiles may influence the design of drug scheduling. Using laboratory evolution in *Enterococcus faecalis*, a gram-positive opportunistic bacterial pathogen typically found in the gastrointestinal tracts of humans [57–62], we measure collateral sensitivity and resistance profiles over time for 20 populations exposed to increasing concentrations of five drugs, yielding 400 strain-antibiotic susceptibility measurement combinations. These collateral profiles reveal a complex story. When data from all drugs are combined, a clear trend emerges: collateral resistance appears more frequently in the early stages of adaptation—when resistance to the selecting drug is lower—while further evolution increasingly favors collateral sensitivity. At the same time, collateral profiles are temporally dynamic and difficult to predict at the level of single drugs or single populations. We show experimentally that optimal drug scheduling may require exploitation of specific time windows where collateral sensitivity is most likely to occur. Finally, using a temporally dynamic stochastic model informed by our experimental measurements, we show that consistent measurements through evolutionary time are required to optimally constrain resistance evolution. Taken together, our results underscore the importance of measuring temporal collateral profiles not only to better understand collateral evolution, but for any future work that hopes to harness collateral effects as a therapeutic option. This work serves to motivate future studies investigating dynamic collateral effects across bacteria species, as well as in cancers, fungi, and viruses.

## Results

### Collateral effects are temporally dynamic

To investigate how collateral effects change over time in *E. faecalis*, we exposed four independent evolutionary replicates of strain V583 to escalating concentrations of a single drug over 8 days (approximately 60 generations) via serial-passage laboratory evolution (Fig 1, Materials and methods). This process was repeated for each drug in Table 1. In a previous study, we characterized collateral effects at the final endpoint of these evolutionary experiments [41]. In this work, we investigate the temporal progression of these collateral effects in a subset of five antibiotics chosen to represent a broad collection of mechanisms of action (Table 1). To

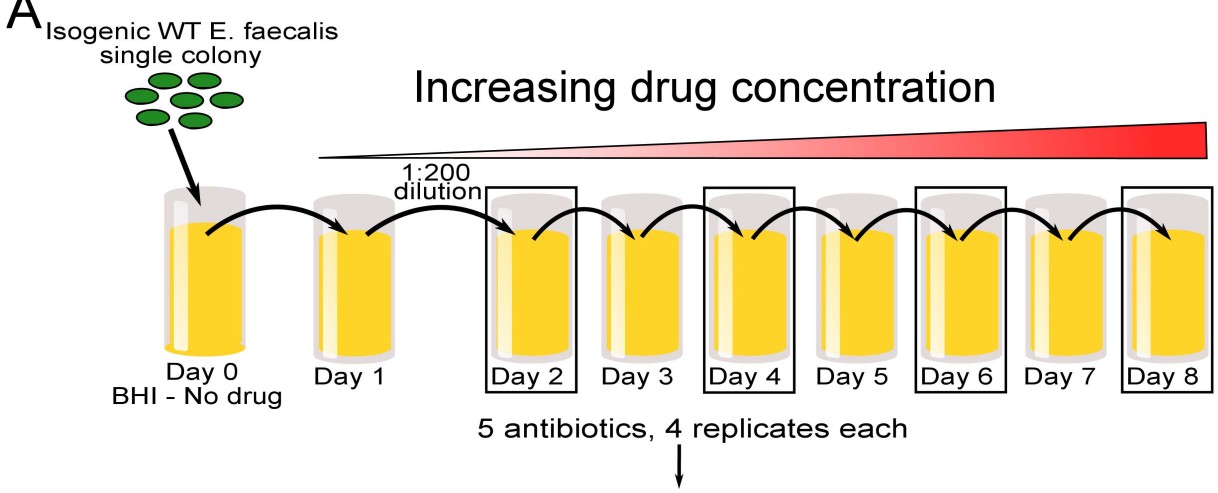

**Fig 1. Laboratory evolution of *E. faecalis* reveals diverse, temporally dynamic collateral sensitivity profiles. (A)** *E. faecalis* strain V583 was exposed to single antibiotics in escalating concentrations over the course of an 8-day serial-passage evolution experiment (roughly 60 total generations). Four independent populations were evolved in the presence of one of five selected antibiotics. The half-maximal inhibitory concentration ($IC_{50}$) was estimated for a single

isolate from each population at days 2, 4, 6, and 8. **(B)** Resistance/sensitivity measurements taken every two days for each of the 20 isolates to each of the 5 antibiotics quantified by the $\log_2$-transformed relative increase in the $IC_{50}$ of the testing drug relative to that of WT V583 cells (black dashed line at $y = 0$). Columns represent the drug used to select the mutant (selecting drugs: Ciprofloxacin (CIP) = red, Daptomycin (DAP) = blue, Doxycycline (DOX) = purple, Ceftriaxone (CRO) = teal, Linezolid (LZD) = orange), whereas rows represent the drug used in the testing assay. Error bars represent the standard error of the mean (SEM) for each individual experiment. Note that the $y$-scales are different for each column but the same within a column to allow for comparison of collateral effects (off-diagonal) with direct effects (diagonal). The data underlying this figure can be found in S1 Data.

**Table 1. Table of antibiotics used in this study and their targets.**

| Drug Name (Abbreviation) | Drug Class | Mechanism of Action |
|---|---|---|
| Ceftriaxone (CRO) | $\beta$-Lactam | Cell wall synthesis inhibitor |
| Ciprofloxacin (CIP) | Quinolone | DNA gyrase inhibitor |
| Daptomycin (DAP) | Lipopeptide | Cell membrane insertion |
| Doxycycline (DOX) | Tetracycline | 30S protein synthesis inhibitor |
| Linezolid (LZD) | Oxazolidinone | 50S protein synthesis inhibitor |

do so, we isolated a single colony from each population at 2-day intervals and measured dose–response curves for each of the antibiotics (Fig 1A).

We quantified resistance and sensitivity by estimating the half-maximum inhibitory dose ($IC_{50}$) for each strain-antibiotic combination (Materials and methods). In total, we estimated the $IC_{50}$ for 400 strain-antibiotic combinations (20 evolving populations made up of 4 independent replicates selected for resistance in the presence of one of the 5 selecting drugs in Table 1, measured against the same 5 antibiotics, at 4 evolutionary time points), each in (technical) replicates of three. For each measurement, we then calculated the collateral response $c \equiv \log_2(IC_{50,Mut}/IC_{50,WT})$, the $\log_2$-scaled fold change in $IC_{50}$ of the evolved strain relative to the ancestral V583 (Fig 1B). Resistance (direct or collateral) corresponds to $c > 0$, while sensitivity corresponds to $c < 0$ ($c = 0$ is indicated by the black dashed line). As in previous work [41,45], we defined collateral resistance or sensitivity to occur when the measured $IC_{50}$ is at least three times the standard error of the mean of the wild-type ($|c| > \pm 3\sigma_{WT}$, where $\sigma_{WT}$ refers to the standard error of the mean in the wild type; see Materials and methods). While other definitions are possible, particularly in cases where a large number of technical replicates are available, we will see (below) that the qualitative trends we observe do not depend sensitively on the threshold for defining sensitivity or resistance.

Our results indicate that resistance to both the drug used for selection and the "unseen" testing drugs varies considerably over time. As expected, resistance to the selecting drug (diagonal entries, Fig 1B) tends to increase approximately (though not exclusively) monotonically over time and, in many cases, plateau after several days of selection. However, the temporal trends in collateral effects—the off-diagonal entries—are variable. In some cases, the population exhibits only collateral resistance (e.g., ciprofloxacin (CIP)-selected populations tested against ceftriaxone (CRO)) at all time points. In other cases, the same population exhibits collateral resistance at one-time point and collateral sensitivity at another (e.g., CRO-selected strains tested against doxycycline (DOX); Linezolid (LZD)-selected strains tested against CRO). Additionally, the variance in outcome across the 4 populations, and over time, depends on the testing drug. For example, resistance to DOX in all isolates varies over a relatively small range (fold-change in $IC_{50}$ varies from a minimum of 0.4 to a maximum of almost 8), while resistance to CRO in the same strains varies substantially more (fold-change in $IC_{50}$ varies from a minimum of 0.007 to a maximum of over 100). We also observed varying levels of induced collateral sensitivity between tested drugs. For example, at no point do any of the 20 evolved strains become collaterally sensitive to LZD, and collateral sensitivity to DAP is rare at all evolutionary time points. This is

particularly notable because LZD and DAP are frequently used as last line of defense antibiotics in the treatment of multi-drug resistant gram-positive infections [63–66].

## Collateral effects are initially dominated by resistance but shift toward sensitivity with further adaptation

To quantify how the propensity for collateral resistance changes over time, we calculated the "instantaneous" collateral resistance (or sensitivity), which measures collateral effects at each time point *relative to the previous time point*, rather than relative to the ancestral strain (Fig 2). For example, to investigate how the population changed between days 4 and 6, we calculate: $c_{inst} \equiv \log_2(IC_{50,D6Mut}/IC_{50,D4Mut})$, where the $IC_{50}$s are calculated at days 6 and 4. This analysis reveals that the first two days of evolution are dominated by collateral resistance effects (91.25%), while the subsequent 6 days of evolution confer collateral resistance at considerably reduced frequencies (33.75%, 38.75%, and 32.50%, respectively). Further, after the first two days of evolution, the last 6 days also share a similar frequency of collateral sensitivity (52.5%, 55.0%, and 57.5%, respectively).

To further quantify these trends, we combined the data from all drugs and all time points and calculated the mean instantaneous collateral effects as a function of resistance to the selecting drug (Fig 2C, left panel; see also Fig A in S1 Text for cumulative collateral effects). At this global level, the mean collateral effects (both instantaneous and cumulative) trend downward—toward sensitivity and away from resistance—as resistance to the selecting drug increases. Notably, the *instantaneous* collateral resistance at early stages (when resistance to the selecting drug is small) is sufficiently large that the mean *cumulative* collateral effects remain positive even at later stages (Fig A in S1 Text, left); that is, the trend toward collateral sensitivity is not strong enough to overcome the initial collateral resistance acquired at low levels of resistance to the selecting drug.

Similarly, the probability (frequency) of collateral sensitivity increases (Fig 2C, right panel; Fig A in S1 Text; $p_{value} < 0.05$ from logistic regression, see Materials and methods) and the probability of collateral resistance decreases (Fig B in S1 Text, $p_{value} < 0.05$) as resistance to the selecting drug is increased. These qualitative trends do not depend sensitively on the specific thresholds used to define CS/CR (Fig C in S1 Text).

One simple explanation of this phenomenon may be an abundance of easily accessible, low-level resistance mutations, perhaps related to efflux pumps [67–69], or a general stress response [70,71], that broadly confer low-level multi-drug resistance at early stages of adaptation. As the antibiotic concentration increases, the population is required to evolve more antibiotic-specific mutations that may be associated with collateral trade-offs.

## Temporally dynamic collateral effects are difficult to predict at the level of individual drugs

While global trends emerge when the data from all drugs is combined, it is not clear whether similar trends occur at the level of individual drugs. To investigate this question, we first calculated correlations between each of the five testing conditions (Fig D in S1 Text) to search for statistical similarities between different drugs. Depending on the metric used, we find that either 4 or 5 of the 10 pairwise combinations produce a statistically significant but relatively weak negative correlation. However, scatter plots indicate that these relationships are nonlinear with considerable scatter, even when statistically significant.

In addition, we found that collateral effects for specific drugs are difficult to predict from collateral profiles measured at earlier times in the same strain. For example, one DAP-adapted population exhibits a significant increase in resistance to CRO between days 2 and 4; however,

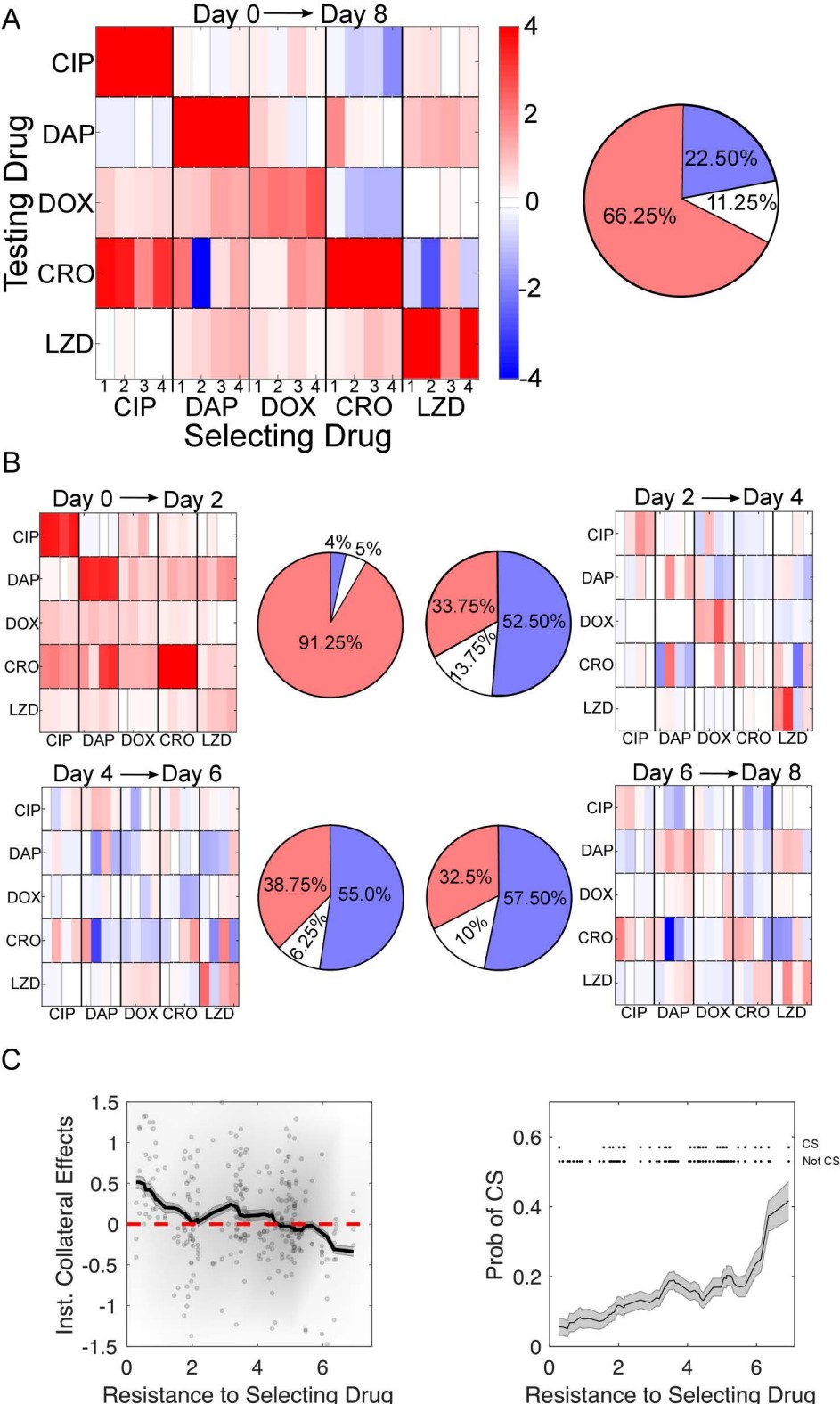

**Fig 2. Early evolution is dominated by collateral resistance followed by a diverse set of collateral responses as evolution continues. (A)** Left Collateral profiles measured at the final time point (Day 8), relative to the day 0 ancestor. Resistance (red) or sensitivity (blue) to each antibiotic is quantified via the $\log_2$-transformed fold change in $IC_{50}$

relative to the ancestral strain (V583). A measurement is deemed to be collaterally resistant or sensitive if it falls above or below three times the standard error of the mean of the wild-type ($|c| > 3\sigma_{WT}$). For each selecting drug, there are four evolutionary replicates (thin rectangular columns). **Right:** Pie chart representing the fraction of collateral effects that confer resistance (red), sensitivity (blue) or no statistical change (white/gray). **(B)** Instantaneous collateral effects: collateral profiles measured for the same 20 mutants as A, but collateral sensitivity/resistance is defined relative to the previous time point (rather than the ancestor strain). Days 0–2, top left; days 2–4, top right; days 4–6, bottom left; days 6–8, bottom right. Over 90% of collateral effects conferred between days 0 and 2 resulted in collateral resistance. The subsequent 6 days of evolution (days 2–4, 4–6, and 6–8) only conferred collateral resistance 34%, 39%, and 33% respectively. **(C)** Instantaneous collateral effects (i.e., collateral effects relative to previous time point) (left) and probability of collateral sensitivity (CS) (right) as a function of resistance to selecting drug. **Left:** dots are individual populations, shading indicates relative density of points, and curve is moving average (shading is ± standard error over each window). **Right:** curve is moving average (shading is ± standard error over each window). Upper inset shows individual data points (top row is CS, bottom row is not CS) used to calculate probability. Averages are performed over windows of size 2. See also Figs A, B, and C in S1 Text. The data and code underlying this Figure can be found in S1 Data and https://doi.org/10.5281/zenodo.14064963.

days 4–6 and 6–8 both come with a significant collateral sensitivity. Similarly, the LZD-selected strains confer resistance to DAP on days 2 and 8, but exhibit collateral sensitivity on days 4 and 6. To quantify this effect, we calculated how often an observation from the current time step correctly predicted change in resistance measurement two days later. That is, if a particular isolate exhibited collateral sensitivity to a drug on day 2, how frequently did it also exhibit collateral sensitivity on day 4? Somewhat surprisingly, collateral profiles on day 2 correctly predict only 41% of day 4 collateral profiles. Similarly, day 4 only successfully predicted 37% of day 6 collateral profiles and day 6 accurately predicted 41% of day 8 profiles. These data indicate that, in contrast to resistance levels to the selecting drug, which tend to be non-decreasing over time, instantaneous collateral effects are largely uncorrelated even between short time intervals (2 days; 10–20 generations) of adaptation. This lack of predictability might also arise if populations are largely heterogeneous over time, meaning that sampling single colonies from each population could result in the appearance of dynamic or stochastic evolution.

## Collateral effects are dynamic within individual populations

To investigate heterogeneity within the evolving populations, we focused on three selection-testing drug pairs (LZD-CRO, CRO-DOX, and CRO-CIP), which correspond to lineages adapted to one of three drugs (LZD, CRO, and CIP) and tested against a different drug (CRO, DOX, and CRO, respectively). In two examples (LZD-CRO and CRO-DOX), the original data suggested collateral resistance to the testing drug decreases over time, with resistance occurring at early stages and sensitivity at later stages. In the third example (CRO-CIP), the collateral effects are approximately constant (resistance) over time.

From each of the three populations, we isolated 48 colonies (12 colonies from each of 4 time points) and measured the response of each isolate to the testing drug using dose–response curves (Fig 3). We found that the specific distributions of $IC_{50}$ values vary in idiosyncratic ways–for example, the LZD-CRO population is characterized by a roughly unimodal distribution whose mean decreases over time, while the CRO-DOX population is characterized by coexistence of highly resistant and highly sensitive isolates at early times, while later time points contain primarily sensitive isolates. Indeed, the mean collateral resistance (i.e., fold change in $IC_{50}$ relative to ancestor) is dynamic—specifically, it is not the same at every time point in the LZD-CRO (one-way ANOVA, $p_{value} = 0.0008$), CRO-DOX ($p_{value} = 0.001$), and CIP-CRO pairs ($p_{value} = 0.05$). See Supporting information for pairwise comparisons at all time points.

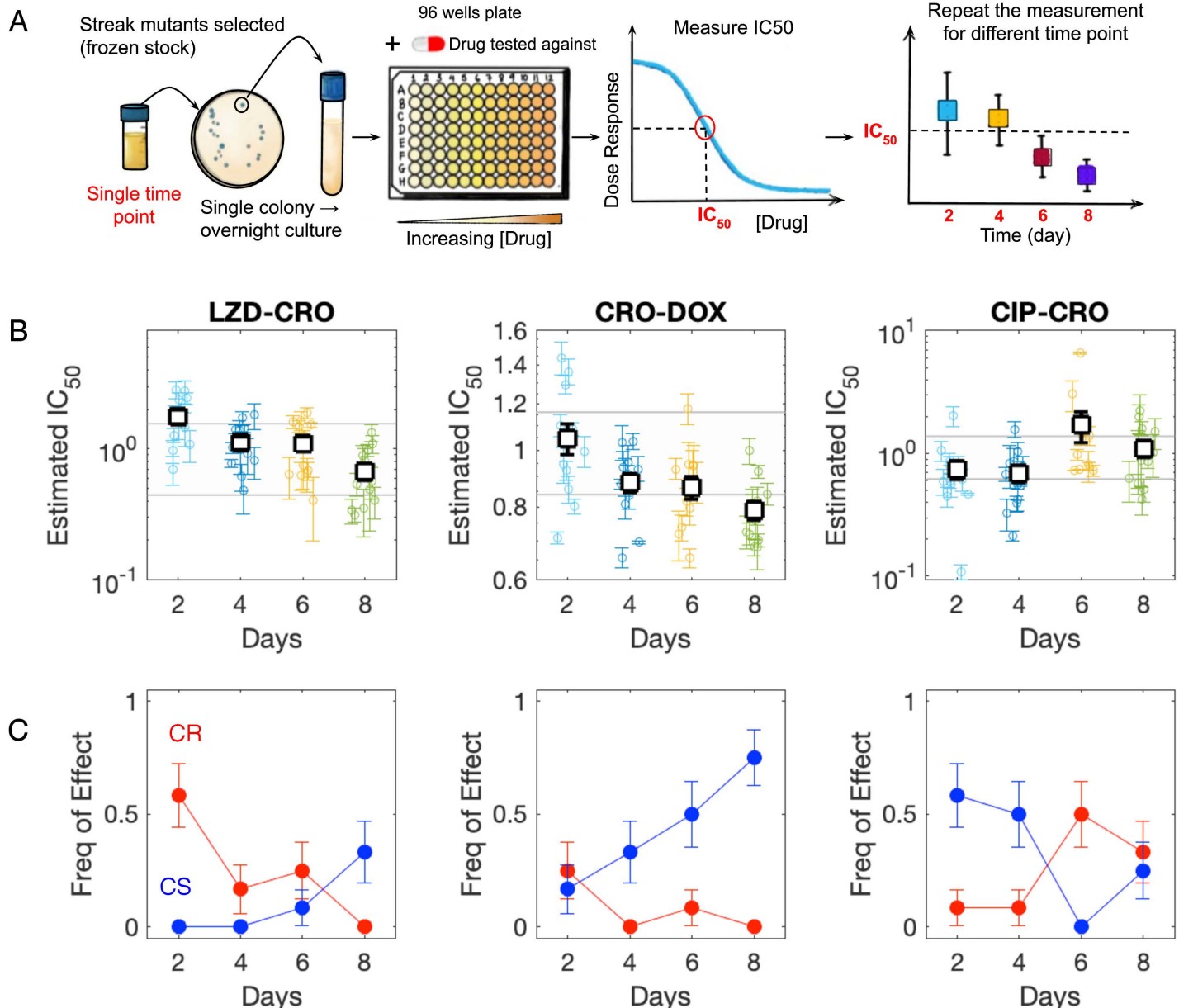

**Fig 3. Collateral effects are dynamic within individual populations.** (A) Samples from a single evolved population at days 2, 4, 6, and 8 are plated and 12 colonies from each time point are selected for phenotyping (IC$_{50}$ characterization). (B) Measured IC$_{50}$ values for 12 isolates at each of four different time points from populations adapted to LZD (left), CRO (middle), and CIP (right) and exposed to CRO (left), DOX (middle), and CRO (right). Shaded region indicates IC$_{50}$ of ancestor strain (±3 standard error). Individual points represent the mean IC$_{50}$ (± standard error from four technical replicates) for a single colony. Open squares represent the population mean—that is, the mean across all colonies (error bars are ± standard deviation of the population). (C) Frequency of colonies exhibiting collateral sensitivity (blue) or collateral resistance (red) over time. See also Fig E in S1 Text. The data and code underlying this Figure can be found in S1 Data and https://doi.org/10.5281/zenodo.14064963.

In addition, we calculated the frequency of CR and CS across isolates. We found that the frequency of sensitive isolates increases over time in the LZD-CRO and CRO-DOX pairs (logistic regression, $p_{value}$ = 0.04 and $p_{value}$ = 0.005, respectively; see Materials and methods) and the frequency of resistant isolates decreases in the LZD-CRO pair ($p_{value}$ = 0.007; for CRO-DOX the trend is qualitatively similar but not significant, $p_{value}$ = 0.1). In contrast to the first

two populations, the third population (CRO-CIP) shows the opposite trend, with resistance increasing in frequency ($p_{value}$ = 0.05) and sensitivity decreasing in frequency over time ($p_{value}$ = 0.02).

These results indicate that the adapting populations exhibit heterogeneity in collateral resistance profiles but nevertheless show clear temporal trends, even within single populations. In some cases (LZD-CRO, CRO-DOX), the trends are reminiscent of the global trends observed across drugs—that is, early stages of adaptation favor collateral resistance and later stages collateral sensitivity—but this is by no means universal (e.g., CRO-CIP exhibits opposite trends).

## Simple model highlights potential trade-offs of multi-drug sequences when collateral profiles are dynamic

Our results raise the question of whether dynamic collateral sensitivity may jeopardize otherwise effective drug sequences, especially those involving multiple drug-switching events. To investigate this question, we developed a simple model (SI) that corresponds to the accumulation of independently combined, but dynamic, collateral effects—intuitively, this could represent a single dominant lineage that sequentially acquires multiple mutations over time. We stress other evolutionary processes are certainly possible, and even likely under some conditions—for example, adaptation may be dominated by a single selection step that enriches for specific mutants from a heterogeneous population. Our goal is not to investigate these dynamics in detail, but simply to provide a null model that quantifies the potential combinatorial patterns associated with the dynamic collateral profiles we measured. In this model, the trends we observe experimentally—namely, the enrichment for resistance at early stages of adaptation—mean that switching between drugs is not always beneficial (Fig F in S1 Text). Instead, switching gives rise to a trade-off: optimal sequences may minimize the resistance to the applied drug (similar to results observed experimentally in [41]), but adding additional drugs tends to raise the global resistance levels—that is, the cumulative resistance of the population to the set of all available drugs. This global resistance does not depend sensitively on the timing of the drugs, though it is precisely the timing that can potentially optimize resistance to the applied drugs.

## Success of switching to a second antibiotic is contingent on temporally dynamic collateral effects

To investigate the effects of drug timing experimentally, we designed an evolution experiment (Fig 4A) meant to approximate a hypothetical drug-switching protocol involving two drugs (CRO and DOX) suggested by simulations to be particularly sensitive to the timing of drug switching (i.e., large difference between best and worst treatments, Fig E in S1 Text). Similar to our previous experiments, we performed serial passage evolution using escalating concentrations of the antibiotic CRO, but now for a total period of 14 days. At the end of each day, we exposed a diluted sample of that population to varying concentrations of a second antibiotic, DOX, to probe how switching to a second drug may have increased or decreased the treatment efficacy. The entire process was repeated for 20 independent populations to quantify evolutionary variability (Fig 4B).

To quantify sensitivity to DOX, we calculated both a global area-under-the-curve sensitivity score, where we take the difference between the area under the dose–response curve of the ancestral strain and each evolved population (Fig 4B, bottom left), as well as the previously described collateral effect metric (i.e., the log-scaled fold change in half-maximal inhibitory concentration). Both metrics show that populations adapted to CRO become increasingly

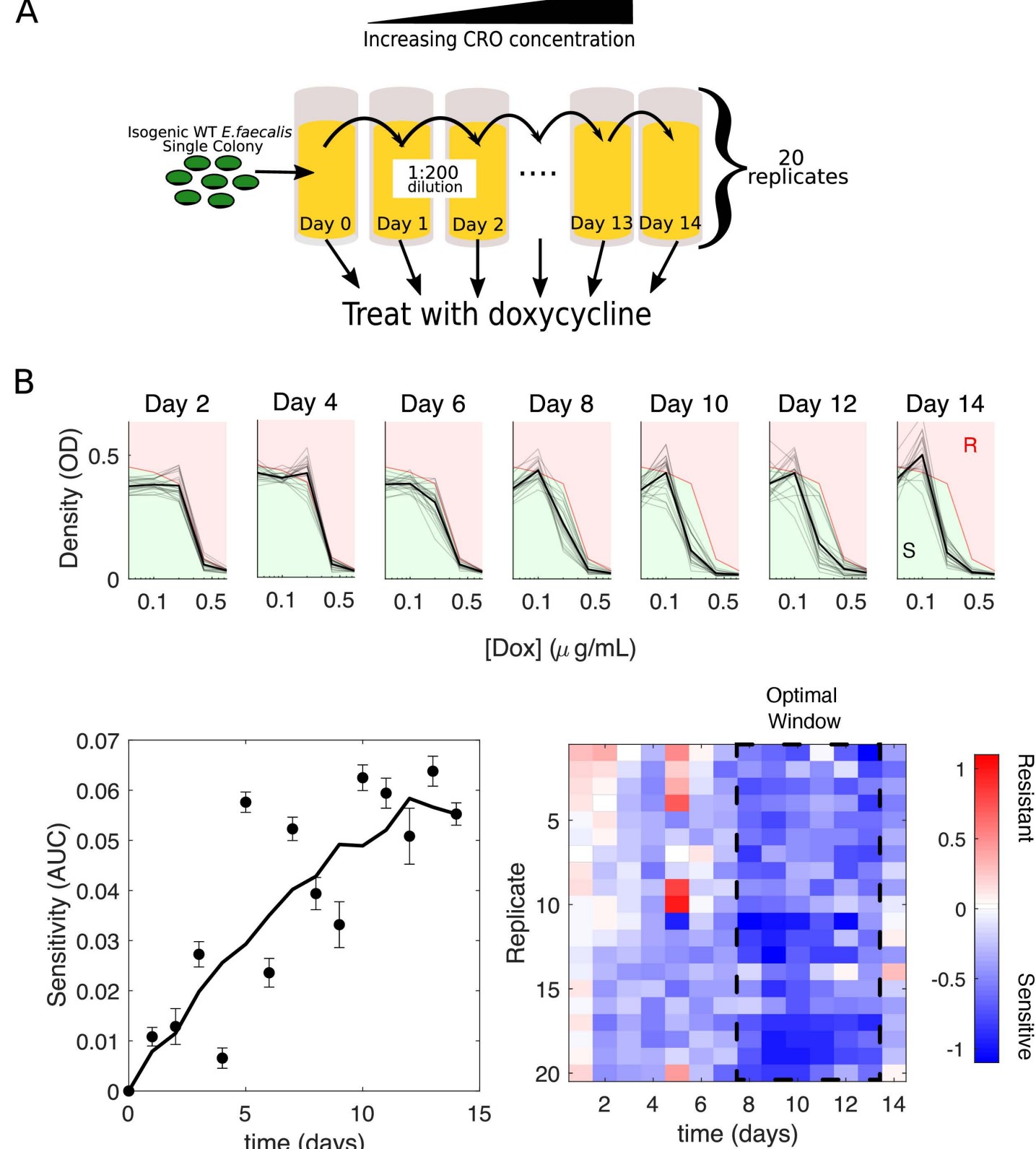

**Fig 4. Success of switching to a second antibiotic is contingent on temporally dynamic collateral effects.** **(A)** Twenty replicate populations of *E. faecalis* V583 were exposed to increasing concentrations of the antibiotic ceftriaxone (CRO) over the course of 14 days via serial passage. At the end of each day, a sample of the

population was isolated and cultured in a range of doxycycline (DOX) concentrations for 24 h. **(B) Top:** Density (OD) after 24 h versus drug (DOX) concentration for all 20 populations (light gray), and the mean over the 20 populations (dark black) at different days of the evolution experiment. For comparison, the dose–response curve in the ancestral strain (red curve) divides the space into regions of increased resistance (red) and increased sensitivity (green). **(B) Bottom left:** Average sensitivity over 20 populations (error bar standard error of the mean) at different time points. Sensitivity is defined as the difference in area under the curve between the ancestral dose–response curve and the dose–response curve of the population in question between the lowest ([Dox] = 0.05 μg/mL) and highest ([Dox] = 0.8 μg/mL) non-zero drug concentrations. Solid line is a moving average. **(B) Bottom right:** Collateral effects (quantified, as before, by $\log_2$-transformed fold change in the $IC_{50}$). Dashed region highlights a transient six-day window where collateral sensitivity is more pronounced. The data and code underlying this Figure can be found in S1 Data and https://doi.org/10.5281/zenodo.14064963.

sensitized to DOX over time. The sensitivity is particularly notable between days 8 and 13, where the mean collateral sensitivity (across replicate populations) is maximized before starting to decline (Fig 4B, bottom right; post hoc pairwise comparisons following one-way ANOVA identify days 8–13 as a statistically distinct cluster). Switching to DOX at time points before or after that window produces different levels of average sensitivity and even leads to resistance in some populations. These results indicate that the effects of a new antibiotic can vary considerably depending on when, along the adaptation trajectory, the new drug is applied.

## Dynamic collateral profiles require dynamic treatments to optimally constrain resistance

Our results indicate that collateral profiles are highly dynamic across evolutionary time. However, due to the stochasticity of evolution and the idiosyncratic nature of our experimental results, it is not clear whether the inclusion of dynamic collateral profiles will result in meaningfully different optimal policies from those that do not account for temporal dynamics. Previously, we developed a simple mathematical model based on a Markov decision process (MDP) to predict optimal drug policies given a set of experimentally measured collateral profiles [41]. Briefly, we begin with a state (resistance profile, initially zeros) and the system stochastically transitions between discrete states. At each time step, we must take an action (choose a drug in our case) and for each state-action pair there is an associated "reward" given. In addition, the action taken influences which future state the system transitions to— the drug chosen for selection impacts the resistance and sensitives that evolve during that step. The output of our model is an optimal policy that maximizes a reward over the given time period. Previously, we used experimental collateral profiles measured after eight days of evolution to construct the transition matrix for our model. Despite encouraging experimental results that showed MDP-based policies outperformed naive two and four drug cycles, this model assumes collateral profiles are static in time.

To address this problem, we extend our model to include temporally dynamic collateral profiles (dynamic Markov Decision Process (d-MDP), see (Materials and methods) Methods). Briefly, the d-MDP solves four static MDPs associated with each of the collateral profiles measured after time periods 0–day 2, day 2–day 4, day 4–day 6, and day 6–day 8. The MDPs are solved in reverse time order, and the cumulative reward function is modified to reward not just the resistance of the currently applied drug, but also evaluate how rewarding the current state will be in subsequent collateral profiles.

For computational efficiency, we discretized both the state space (resistance to each drug can only take integer values between −5 and 25) as well as the experimentally measured collateral profiles (antibiotic exposure can lead to increases/decreases of −2, −1, 0, 1, or 2). We have previously shown qualitatively similar results for different discretization schemes [41].

Using evolutionary simulations, we compare the d-MDP policy (TD-opt) to four static MDP policies derived from collateral profiles measured after day 2, day 4, day 6, and day

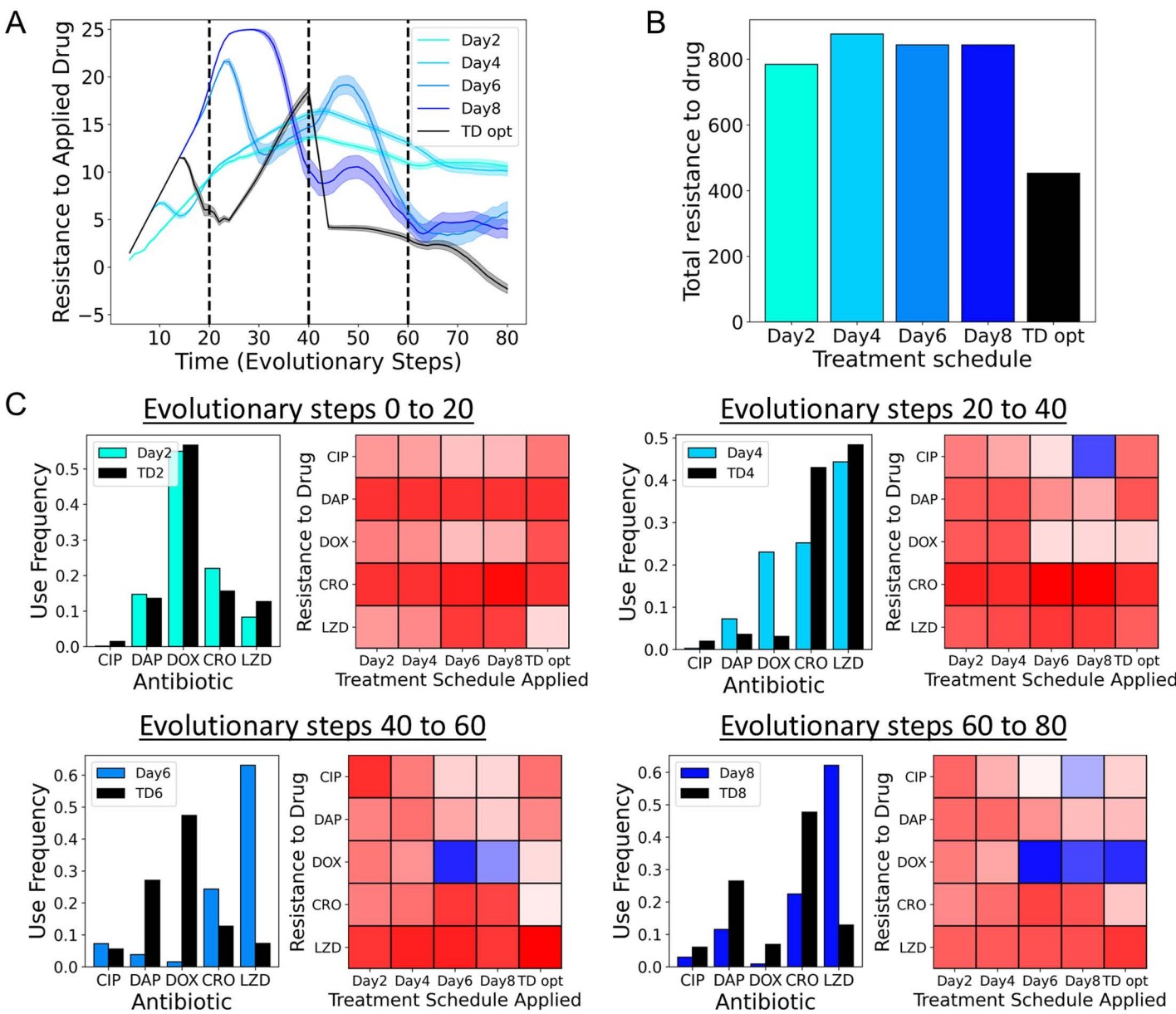

**Fig 5. Dynamic collateral profiles require dynamic optimal treatments.** (A) Average level of resistance ($\langle R(t) \rangle$) to the applied drug for MDP policies calculated from collateral profiles measured after two days (teal), four days (light blue), six days (blue), eight days (dark blue). Time-dependent MDP optimal with knowledge of the dynamic collateral profiles is shown in black. Resistance to each drug is characterized by 31 discrete levels arbitrarily labeled with integer values from −5 (least resistant) to 25 (most resistant). At time 0, the population starts with 0 resistance to all drugs. Evolutionary dynamics for evolutionary steps 0–20, 20–40, 40–60, and 60–80 correspond to the experimental profiles measured between days 0–2, days 2–4, days 4–6, and days 6–8, respectively. (B) Total resistance to the applied drug (area under treatment curves in **A**) for each simulated treatment schedule. (C) **Bar charts:** For each regime (set of 20 evolutionary steps) the time-dependent MDP optimal policy is compared to the corresponding MDP policy calculated from experimental collateral profiles exclusively at that time. (C) **heatmaps:** Average resistance profile at the end of that evolutionary period for each of the treatment schedules applied. Dark red corresponds to high resistance, while dark blue corresponds to sensitivity. The data and code underlying this Figure can be found in S1 Data and https://doi.org/10.5281/zenodo.14064963.

8 (Fig 5). Crucially, the simulations reflect the dynamic nature of the measured collateral profiles and consist of four segments each associated with the corresponding collateral profiles (dynamics of evolutionary steps 0–20 are derived from the collateral profiles measured

between days 0 and 2, evolutionary steps 20–40 are derived from days 2 to 4, and so on). In comparison to the static MDP solutions, we find that TD-opt both better constrains long-term resistance (Fig 5A) and leads to lower total resistance (Fig 5B, 50% lower total resistance). Crucially, the d-MDP policy incorporates the reward of future collateral profiles allowing for accurate cost-benefit analysis of each action in the context of dynamic collateral profiles. In addition, the experimental data shows the distribution of collateral effects is not constant with evolutionary time. The d-MDP understands which collateral effects occurred over each time period, whereas the static policies are the combined effects of multiple changes.

We can see how different the policies are by comparing the antibiotic frequencies of the static MDP policy derived from experimental measurements at the end of that period, with the TD-opt policy over that same period (Fig 5C, bar charts). We see that over time the policies heavily diverge (day 2 optimal versus TD2 is very similar, while day 6 and day 8 optimal profiles show large differences in policy with the TD6/TD8 counterparts), reinforcing the importance of measuring dynamic collateral profiles. Finally, at the end of each 20-step segment, we compared the average resistance profile under each policy (Fig 5C, heatmaps). Strikingly, we find that the static MDP policies fail despite creating transient periods of collateral sensitivity. For example, the static MDP policies on day 6 and day 8 can result in extremely sensitive states. However, because these sensitivities are not predicted by the static models, these policies fail to exploit them for long-term control. By contrast, the d-MDP policy predictably creates, and optimally leverages, these sensitive states, thereby better constraining resistance.

## Discussion

Our work provides systematic evidence of temporal collateral drug effects in the pathogen *E. faecalis*. Broadly, we observe an initial strong bias toward collateral resistance, while subsequent evolutionary steps are more balanced between collateral sensitivity and resistance relative to their most recent evolutionary isolate. Often, the initial acquisition of collateral resistance is of sufficient magnitude to counter future relative sensitivity gains, resulting in a less susceptible population relative to the original ancestor. This conclusion is likely dependent on the length of the overall evolutionary experiment, where further evolution may ultimately fully reverse the initial resistance acquisition; however, additional studies are required. The time-dependent nature of collateral sensitivity and collateral resistance presents both additional challenges and new opportunities for designing multi-drug therapies to slow resistance. This idea is underscored by our 14-day evolution experiment that highlights the potential transience of collateral sensitivity windows we seek to leverage in treatment (Fig 4B), in addition to our d-MDP results, which reveal just how sensitive optimal treatments are to dynamic underlying collateral profiles (Fig 5). Full optimization of sequential drug therapies will likely involve not merely static end-point measurements of collateral effects, but a full description of their temporal development.

The goal of this study was to broadly survey phenotypic resistance patterns over time in a systematic way. This approach comes with obvious drawbacks, and we are left with many unanswered questions. Most notably, our work does not provide any information about the molecular mechanisms underlying the collateral effects; such insight, while hard won even for a single pair of drugs, will be essential to fully exploit the phenotypic effects observed here. In addition, we focused on a single bacterial species, and all experiments were performed starting from the ancestral V583 *E. faecalis* strain. Our results complement several recent studies on collateral effects that suggest sequential therapy may help slow or reverse resistance in bacteria [10,31,41], fungi [72], and cancers [38]. However, sequential therapy has led to mixed results when tested [36,73]. This work suggests that collateral profiles are themselves evolving,

providing additional context as to why sequential therapy may not provide the expected treatment benefits. It is not clear how the results might change in a different strain or species, though data from several recent studies suggest the presence of dynamic features in collateral profiles in other species [10,36,52,54]. Indeed, recent work underscores just how important genotype can be in antibiotic evolvability [74], and the search for more general patterns is ongoing [36,52,54]. It is also possible that at least some of the trends we observe are specific to the precise laboratory evolution protocol used here, where drug is systematically increased over time following serial dilution. Future work may aim to identify how these patterns change in cells from different physiological states (e.g., biofilm [75]) or under different modes of selection pressure [76].

Finally, it is important to keep in mind the scope of our work. These evolution experiments are done in a highly controlled laboratory environment. Our protocols are not meant to guide clinicians, but instead focus on whether or not collateral profiles change even in the simplest of drug-evolution environments—free, for example, of the important but difficult-to-quantify interactions between host and pathogen. Translating these results into a clinically accurate model would require additional work to understand the mechanistic, clinical, and even theoretical principles governing drug sequence optimization. This work serves as a reminder of the complexities of evolution and the still long path we must walk to confidently prescribe effective dosing schedules in patients. At the same time, the results highlight the rich dynamical behavior of collateral sensitivity in even simplified laboratory populations, offering a largely unexplored frontier for evolution-based control strategies.

# Materials and methods

## Strains, antibiotics, and media

All resistance evolution lineages were derived from an *E. faecalis* V583 ancestor, a fully sequenced clinical isolate with vancomycin resistance [77]. The 5 antibiotics used in this study and their mechanisms of action are listed in Table 1. Antibiotics were prepared from powder stock and stored at an appropriate temperature. Evolution and $IC_{50}$ measurements were conducted in brain heart infusion (BHI).

## Laboratory evolution experiments

Evolution experiments were performed in replicates of four. Daily serial passage evolutions were conducted in 1 mL BHI medium in 96-well plates with a maximum volume of 2 mL. Each day populations were grown in three antibiotic concentrations spanning sub- and super-MIC doses. After approximately 16 of incubation at 37 °C, the well with the highest drug concentration that contained visible growth was propagated into three new concentrations (typically one-half, 2×, and 4× the highest concentration that had visible growth). A 1/200 dilution was used as an inoculum for the next day's evolution plate. This process was repeated for 8 days for the multi-drug study and 14 days for the CRO/DOX study. All strains were stocked in 30% glycerol and subsequently plated on pure BHI plates for further experimentation. A single colony was selected for $IC_{50}$ determination. To help ensure no contamination occurred, cells were regularly plated and visualized using DIC microscopy to ensure *E. faecalis* morphology.

## Measuring drug resistance and sensitivity

$IC_{50}$ determination experiments were performed in 96-well plates by exposing each strain to a drug gradient consisting of between 6 and 14 concentrations, typically in linear dilution series prepared in BHI medium with a total volume of 205 μL (200 μL BHI, 5 μL of 1.5 OD cells) in each well. After 20 h, we measured the OD at 600 nm via an Enspire Multi-modal Plate Reader

(Perkin Elmer) with an automated plate stacker. OD measurements for each drug were normalized by the OD600 in the absence of drug.

In order to quantify resistance to each drug, the OD600-generated dose–response curve was fit to a Hill-like function $f(x) = \left[1 + \left(\dfrac{x}{K}\right)^h\right]^{-1}$ using a non-linear least square fitting.

$K$ is the $IC_{50}$ and $h$ is a Hill coefficient that represents the steepness of the dose–response curve. Strains were deemed "collaterally resistant" or "collaterally sensitive" if its $IC_{50}$ had increased or decreased by more than three times the standard error of the wild-type mean $IC_{50}$. In previous work, we chose this threshold because it minimized the number of false positives (e.g., all measurements of $IC_{50}$ in the ancestor strains, across all drugs, fell within this $\pm 3\sigma_{WT}$ window) [41,45]. All dose–response curves were measured in technical replicates of 3 or 4.

## Logistic regressions for estimating frequency trends

To characterize trends in the frequency of resistance (CR) or sensitivity (CS), we used standard logistic regression. Specifically, we assume the classic logit model where $\dfrac{\ln p}{(1-p)} = c_1 + c_2 X$, with $p$ the probability of CR (or CS), $c_1$ and $c_2$ regression coefficients, and $X$ the predictor variable of interest (either resistance to the selecting drug, as in Fig 2, or time, as in Fig 3. An increasing trend corresponds to $c_2 > 0$ and a decreasing trend to $c_2 < 0$. P-values ($p_{value}$) are given for each regression in the corresponding figure caption.

## MDP models

The MDP models consist of a finite set of states ($S$, resistance profile), a finite set of actions ($A$, antibiotics), a conditional probability ($P_a(s'|s,a)$) describing (action-dependent) Markovian transitions between these states, and an instantaneous reward function ($R_a(s)$) associated with each state and action combination. The state of the system $s \in S$ is an $n_d$-dimensional vector, with $n_d$ indicating the number of drugs and each component $s^i \in \{r_{min}, r_{min} + 1, \ldots, r_{max}\}$ indicating the level of resistance to antibiotic $i$. The action $a \in A \equiv \{1, 2, \ldots, n_d\}$ is the choice of antibiotic at the current step.

In previous work, we considered a system described by a static collateral profile derived from experimental measurements after eight days of evolution. In the non-dynamic MDP, we take the reward function $R_a(s)$ to be the (negative of the) resistance level to the currently applied drug (i.e., the $a$-th component of $s$). The optimal policy is chosen to maximize a cumulative reward function $R_c = \sum_{t=0}^{\infty} \gamma^t \langle R_\pi(s_t) \rangle$, where $t$ is the time step, $s_t$ is the state of the system at time $t$, $R_\pi(s_t)$ is a random variable describing the instantaneous reward assuming that the actions are chosen according to policy $\pi$, and brackets indicate an expectation value. The parameter $\gamma$ ($0 \leq \gamma < 1$) is a discount factor that determines the relative importance of instantaneous versus long-term optimization. Each MDP was solved using value iteration, a standard dynamic programming algorithm used in MDP models. Once the optimal policy is found, the system is reduced to a Markov chain with transition matrix $T_{\pi^*} = P_{\pi^*(s)}\left(s'|s, \pi^*(s)\right)$, where the subscript $\pi^*$ means that the decision in each state is determined by the policy $\pi^*$ (i.e., that $a = \pi^*(s)$ for a system in state $s$).

In this work, we consider a system described by a dynamic collateral profile derived from experimental measurements every two days of evolution (days 2, 4, 6, and 8). Initially, one might apply the previous MDP model on each of the four static collateral profiles independently and subsequently apply the day 2 optimal initially, followed by the day 4 optimal, and so on. However, this solution fails to converge to a true long-term optimal. This is

because the day 2 optimal has no information on the collateral profiles (or transition matrix) that describe the day 2–day 4 evolution and as a result may leave the system in a non-optimal state to begin the day 4–day 6 evolution. In the most extreme case, a policy may be optimal for that transient evolutionary period, but leave the system trapped in a high resistance state space as the collateral profiles dynamically switch.

To combat this, there are two important changes. First, we must include in our cumulative reward function a reward that is connected to the value of that state in future collateral profiles, not just the current collateral profile. However, for this to be accurate, we must know what the future reward values are. Therefore, the second change is to calculate the time-dependent optimal in reverse order, beginning with the day 6–day 8 period. The d-MDP solution can be outlined as follows:

1. Using the static MDP, calculate the optimal policy using the final evolutionary collateral profile (day 6–day 8). The instantaneous reward is the (negative of the) resistance level to the currently applied drug.

2. Exhaustively simulate the evolutionary outcome of each initial state using the optimal policy calculated in step 1. This gives us a reward value that is associated with entering the final evolutionary period in that state.

3. Calculate the optimal policy using the day 4–day 6 collateral profiles. The instantaneous reward is now a combination of the (negative of the) resistance level to the currently applied drug and how rewarding that state is to be in, given our simulations performed in step 2 (Note: this balance between current resistance and future state success can be modified to fit the needs of the user and is a trade-off between lower average resistance and lower final resistance).

4. Similar to step 2, exhaustively simulate the evolutionary outcome of each initial state using the optimal policy calculated in step 3. This is our reward value associated with entering the day 4–day 6 period in that state.

5. Calculate the optimal policy using the day 2–day 4 collateral profiles. Again, the reward is a combination of resistance level to the applied drug and future state reward as calculated by the simulations in step 4.

6. Repeat the two-step process above until you reach the initial time period.

Finally, we note that this process can be generalized to any discreditable set of collateral profiles or desired reward objective. For example, if evolution happens on a faster timescale, one can add more frequent experimental collateral profile measurements. If one prefers to minimize the total resistance state of the system, one can do that, rather than focus on maintaining sensitivity to at least one drug as we do here. More importantly, we propose a model capable of calculating optimal policies to reach some objective, given a stochastic evolving system with temporally dynamic transition matrices.

## Supporting information

**S1 Data. Supplemental raw data.**
(EXCEL)

**S1 Text. Supplementary text.** Fig A. Cumulative collateral effects exhibit early resistance but trend toward sensitivity with adaptation. Fig B. Collateral resistance decreases as resistance to selecting drug increases. Fig C. Modulating threshold value does not change qualitative features of collateral effects. Fig D. Correlation between collateral effects between different

testing antibiotics. Fig E. Frequency trends within individual populations do not depend sensitively on threshold criteria for defining collateral effects changes. Fig F. Simulations suggest optimal dosing can be aided or hindered by temporally resolved collateral profiles.
(PDF)

## Author contributions

**Conceptualization:** Jeff Maltas, Anh Huynh, Kevin B. Wood.

**Data curation:** Jeff Maltas, Anh Huynh.

**Formal analysis:** Jeff Maltas, Anh Huynh, Kevin B. Wood.

**Funding acquisition:** Kevin B. Wood.

**Investigation:** Jeff Maltas, Anh Huynh, Kevin B. Wood.

**Methodology:** Jeff Maltas, Kevin B. Wood.

**Supervision:** Jeff Maltas.

**Validation:** Jeff Maltas, Anh Huynh.

**Visualization:** Jeff Maltas, Kevin B. Wood.

**Writing – original draft:** Jeff Maltas, Kevin B. Wood.

**Writing – review & editing:** Jeff Maltas, Anh Huynh, Kevin B. Wood.

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
