## [Editor Report · Decision Letter 0]

12 Aug 2024

Dear Dr Maltas,

Thank you for submitting your manuscript entitled "Dynamic collateral sensitivity profiles highlight challenges and opportunities for optimizing antibiotic sequences" for consideration as a Update Article by PLOS Biology.

Your manuscript has now been evaluated by the PLOS Biology editorial staff, as well as by an academic editor with relevant expertise. After discussion with the previous Academic Editor, we would like to send your new submission to review inviting the same reviewers as in the previous submission.

Once your full submission is complete, your paper will undergo a series of checks in preparation for peer review. After your manuscript has passed the checks it will be sent out for review. To provide the metadata for your submission, please Login to Editorial Manager (https://www.editorialmanager.com/pbiology) within two working days, i.e. by Aug 14 2024 11:59PM.

Kind regards,

Melissa

Melissa Vazquez Hernandez, Ph.D.

Associate Editor

PLOS Biology

---

## [Decision Letter · Decision Letter 1]

11 Oct 2024

Dear Jeff,

First of all in name of our Editor in Chief and the whole PLOS Biology team, we are deeply sorry for the loss of Dr. Wood and we would like to extend our sympathies and condolences to you, his team and his family.

Thank you so much for your patience while we considered your revised manuscript "Dynamic collateral sensitivity profiles highlight challenges and opportunities for optimizing antibiotic sequences" for publication as a Update Article at PLOS Biology. This revised version of your manuscript has been evaluated by the PLOS Biology editors, the Academic Editor and most of the original reviewers.

Based on the reviews and on our Academic Editor's assessment of your revision, we are likely to accept this manuscript for publication, provided you satisfactorily address the remaining points raised by the reviewers. I would also like to point out that the Academic Editor requested that the figures were adjusted since the font is really small. Please also make sure to address the following data and other policy-related requests.

a) We routinely suggest changes to titles to ensure maximum accessibility for a broad, non-specialist readership, and to ensure they reflect the contents of the paper. In this case, we would suggest a minor edit to the title, as follows. Please ensure you change both the manuscript file and the online submission system, as they need to match for final acceptance:

"Dynamic collateral sensitivity profiles highlight opportunities and challenges for optimizing antibiotic treatments"

Please supply the numerical values either in the a supplementary file or as a permanent DOI’d deposition for the following figures:

Figure 1B, 2ABC, 3BC, 4B, 5ABC, S1, S2, S4, S4, S5, S6ABCD

c) Please cite the location of the data clearly in all relevant main and supplementary Figure legends, e.g. “The data underlying this Figure can be found in S1 Data” or “The data underlying this Figure can be found in https://doi.org/10.5281/zenodo.XXXXX”

d) Please ensure that your Data Statement in the submission system accurately describes where your data can be found and is in final format, as it will be published as written there.

e) Per journal policy, if you have generated any custom code during the course of this investigation, please make it available without restrictions upon publication. Please ensure that the code is sufficiently well documented and reusable, and that your Data Statement in the Editorial Manager submission system accurately describes where your code can be found.

f) Please note that per journal policy, the complete model system/species (Enterococcus faecalis) studied should be clearly stated in the abstract of your manuscript.

g) As sad and unfortunate as his passing is, please indicate in the revise version that Dr. Wood is deseased so the operations team can activate a specific protocol for this cases.

We expect to receive your revised manuscript within two weeks.

*Published Peer Review History*

*Press*

Sincerely,

Melissa

Melissa Vazquez Hernandez, Ph.D.

Associate Editor

PLOS Biology

REVIEWERS' COMMENTS

Reviewer #1:

Maltas et al submitted a revised version on dynamic collateral sensitivity profiles in an Enterococcus model. I remain very enthusiastic about the manuscript, which the authors again improved substantially. The finding of dynamic collateral effects during the evolution of resistance is an important novel finding, previously not properly documented. The finding is well supported by evolving the model bacterium to high resistance against 5 different antibiotics, combined with comprehensive resistance testing. The initial insights obtained are further corroborated by two types of mathematical models and, in my opinion most important, additional follow-up experiments, including one where the dynamic nature of collateral sensitivity and its potential to constrain bacterial adaptation is evaluated experimentally (see Fig. 4).

I would like to add that in my opinion, the authors have very well addressed the concerns by reviewers. I particularly like the addition of the new model based on dynamic MDP. I also agree with the authors reply on the validity of the findings presented in Figures 1 and 2, and very much welcome the addition of further statistical analyses to support the conclusions drawn. I also agree with the authors that it is very well justified to focus and highlight both global trends as well as individual dynamics in particular populations. In my opinion, the identification of global trends is of particular value, because it highlights prevailing processes that apply across different antibiotics. This manuscript represents a major advance in understanding, especially regarding the here repeatedly shown dynamics of collateral effects. Therefore, this manuscript is very well suited for publication in PLoS Biology.

I still have a few minor suggestions for improving the manuscript:

1) The authors could emphasize in the introduction and at the beginning of the discussion that Enterococcus faecalis is used as a model to assess the dynamics of collateral effects, thereby making clear that future work in other bacterial taxa is important to further explore the findings made.

2) Third last sentence of the abstract: It should be "decreases in isolates.." instead of "..decreases is isolates.."

3) Last sentence of abstract: I would mention and thus highlight already in the abstract that the mathematical model is based on dynamic Markov Decision Process.

4) Line 31: Gram should be written with a Capital G

5) Line 49: I recommend to emphasize that evolution experiments were performed with each of 5 antibiotics listed in table 1. The sentence in lines 47-50 may be mis-read that only 1 antibiotic was used for the evolution experiments.

6) Line 58: A related point: I would explain that the 20 evolving populations are 4 independent replicate populations evolving in the presence of either of 5 antibiotics, in order to avoid any misunderstanding (such that only a single antibiotic was used for these evolution experiments).

7) Legend to figure 1: I would replace "mutants" with "isolates"

8) Legend to figure 1: I have the impression that the description of rows and columns is mixed up. The rows show the testing drugs and the columns the selecting drugs. If not, then the labels in the panel should be switched.

9) At the beginning of the discussion, I recommend to add that there is a dynamic RELATIVE increase in collateral sensitivity over time, while in comparison with the ancestor cross-resistance dominates.

10) In the discussion, I would highlight more clearly the very insightful validation experiment shown in figure 4 and also the new model based on dynamic MDP. Both are a particular strength and should thus be more visible.

Reviewer #2:

I was supportive of the prior version and appreciate this version more, especially with the added stochastic control model to evaluate effects of time dependency on resistance constraint. I find the work well written and valuable. My broader editorial recommendation is that space and time permitting, the paper would benefit from an additional paragraph in the Discussion that places these studies in context of other work, both on modeling and testing collateral sensitivity evolution in Ef and in biological models with other pathogens or cancers. This would help to connect this study with the field more concretely.

---

## [Editor Report · Decision Letter 2]

5 Dec 2024

Dear Jeff,

Thank you for the submission of your revised Update Article "Dynamic collateral sensitivity profiles highlight opportunities and challenges for optimizing antibiotic treatments" for publication in PLOS Biology. On behalf of my colleagues and the Academic Editor, Tobias Bollenbach, I am pleased to say that we can in principle accept your manuscript for publication, provided you address any remaining formatting and reporting issues. These will be detailed in an email you should receive within 2-3 business days from our colleagues in the journal operations team; no action is required from you until then. Please note that we will not be able to formally accept your manuscript and schedule it for publication until you have completed any requested changes.

IMPORTANT: I really appreciate all the work behind in the revision, I would just like to ask you one last thing if possible. In the previous decision letter I ask to modify the font size of the figures since they were so small. I believe this was not modify. If possible, could you please modify them before passing them to production? I can of course understand if this is not possible, I just think it would be nicer for the reader.

PRESS

Sincerely, 

Melissa Vazquez Hernandez, Ph.D., Ph.D.

Associate Editor

PLOS Biology
